# Clinical Trials for Use of Melatonin to Fight against COVID-19 Are Urgently Needed

**DOI:** 10.3390/nu12092561

**Published:** 2020-08-24

**Authors:** Konrad Kleszczyński, Andrzej T. Slominski, Kerstin Steinbrink, Russel J. Reiter

**Affiliations:** 1Department of Dermatology, University of Münster, Von-Esmarch-Str. 58, 48149 Münster, Germany; Kerstin.Steinbrink@ukmuenster.de; 2Department of Dermatology, Comprehensive Cancer Center, University of Alabama at Birmingham, Birmingham, AL 35294, USA; aslominski@uabmc.edu; 3Pathology and Laboratory Medicine Service, VA Medical Center, Birmingham, AL 35294, USA; 4Department of Cellular and Structural Biology, UT Health, San Antonio, TX 78229, USA; reiter@uthscsa.edu

**Keywords:** melatonin, COVID-19, inflammation, immune response, clinical trials

## Abstract

The recent pandemic of COVID-19 has already infected millions of individuals and has resulted in the death of hundreds of thousands worldwide. Based on clinical features, pathology, and the pathogenesis of respiratory disorders induced by this and other highly homogenous coronaviruses, the evidence suggests that excessive inflammation, oxidation, and an exaggerated immune response contribute to COVID-19 pathology; these are caused by severe acute respiratory syndrome coronavirus 2 (SARS-CoV-2). This leads to a cytokine storm and subsequent progression triggering acute lung injury (ALI)/acute respiratory distress syndrome (ARDS), and often death. We and others have reported melatonin to be an anti-inflammatory and anti-oxidative molecule with a high safety profile. It is effective in critical care patients by reducing their vascular permeability and anxiety, inducing sedation, and improving their quality of sleep. As melatonin shows no harmful adverse effects in humans, it is imperative to introduce this indoleamine into clinical trials where it might be beneficial for better clinical outcomes as an adjuvant treatment of COVID-19-infected patients. Herein, we strongly encourage health care professionals to test the potential of melatonin for targeting the COVID-19 pandemic. This is urgent, since there is no reliable treatment for this devastating disease.

## 1. Introduction

As of today (22 July 2020) there have been more than 616,317 deaths worldwide from coronavirus (COVID-19), a newly emerged respiratory disease caused by severe acute respiratory syndrome coronavirus 2 (SARS-CoV-2). Regarding the most affected countries to date, the USA has reported 142,066 deaths, Brazil 81,487, the United Kingdom 45,422, Mexico 40,400, Italy 35,073, and France 30,165 [1]. These large numbers warrant urgent research to accelerate clinical trials with therapies that may reduce the alarmingly high death rate. The combined use of anti-viral and anti-inflammatory drugs may be more efficient than using either modality alone. One of the overlooked and promising candidates is melatonin, which may substantially enhance the actions of adjuvant treatments for COVID-19 by reducing symptoms such as pneumonia, acute lung injury (ALI), and acute respiratory distress syndrome (ARDS). At present, the low efficacy of anti-viral drugs on COVID-19 is not surprising. Due to increased drug resistance and continuously occurring mutations of the virus, we still lack ideal medicines to target this disease and new vaccines have to be repeatedly adapted to the continuously changing viral subtypes. In fact, the drugs in the market can only mitigate the mild to moderate symptoms if used in the early stage of viral infection, and have reduced effects in patients with severe symptoms or those who are predisposed to complications. Thus, the clinical significance is limited, as well as because viral infectious diseases are self-limiting and the mild to moderately severe patients develop self-recovery without treatment. In viral infectious diseases, the key is to ameliorate the severe symptoms, including the massive tissue and organ injury and, finally, to control the mortality. It has been speculated that the most severe symptoms are beyond viral cytotoxicity *per se* and result from the overreaction of the innate immune response that causes destructive inflammation, as observed in the severe disease progression of coronavirus infections [2]. This may be one of the reasons why antiviral drugs have failed to be effective in severely infected patients. To compensate for the shortcomings of the anti-viral drugs, a more generalized and less virus-specific therapy which instead targets severe symptoms of the viral infection should be considered. Melatonin is a suitable candidate. Melatonin possesses an excellent anti-oxidative and anti-inflammatory capacity and it balances the overshooting innate immune response while promoting adaptive immunity [3,4,5,6]. Currently, an increasing number of publications has suggested or strongly recommended the use of melatonin to combat COVID-19.

## 2. Pathogenesis of COVID-19

To date, the effect of SARS-CoV-2 on humans have been clearly age-related. Thus, the excessive mortality rate occurs in the elderly with very few deaths from COVID-19 being recorded for individuals under the age of 20. Currently reported COVID-19-affected patients present varying symptoms including fever, dry cough, myalgia, fatigue, or diarrhea. In other cases, the acute progression of the disease results in ALI/ARDS, respiratory failure, heart failure, sepsis, and sudden cardiac arrest within a few days [7,8,9]. The pathogenic examination of lungs from mild COVID-19 patients revealed edema, proteinaceous exudate with globules, patchy inflammatory cellular infiltration, and moderate formation of hyaline membranes [10]. In a postmortem assessment of a COVID-19 patient with severe ARDS, specimens of the infected lungs demonstrated bilateral diffuse alveolar damage with edema, pneumocyte desquamation, and hyaline membrane formation [11]. Although these reports were performed for only a small number of cases, they do resemble the pathological features identified in SARS- or MERS-induced respiratory disorders [12].

As recently reviewed [13], SARS-CoV-2 shares 79.0% nucleotide identity with SARS-CoV and 51.8% identity with MERS-CoV, indicating their high genetic homology. In SARS-CoV and MERS-CoV, infected animal models revealed an inflammatory response which causes a “cytokine storm”, subsequently triggering vascular leakage and abnormal innate and adaptive immune responses, including lymphopenia and an increase in neutrophils, thereby inducing ALI/ARDS or even death [14]. In the early stages of coronavirus infections, dendritic and epithelial cells are activated and cause a reported “deluge” of pro-inflammatory cytokines, i.e., elevated levels of interleukin-1β (IL-1β), IL-6, interferon-γ (IFN-γ), interferon-inducible protein 10 (IP-10) or IL-4, IL-10, and IL-17 [5,9]. It was documented that repressed immune functions in COVID-19 patients are accompanied by lymphopenia and neutropenia, as well as a decreased number of CD8+ T cells [7,8,9]. Furthermore, recent reports suggest that some COVID-19 patients, although negative for the viral nucleic acid assay, still sometimes present with a high level of inflammation. Altogether, the most recent findings indicate that inflammation is a major issue for COVID-19 patients in whom the immune system is severely attenuated due to the high cytokine production that contributes to the COVID-19 pathogenesis. It should be added that the amplification of the inflammatory response would promote programmed cell death (apoptosis) or necrosis of the affected cells, which would further trigger inflammation, followed by the increasing permeability of blood vessels and the aberrant accumulation of inflammatory cells, including monocytes, macrophages, and neutrophils. The resultant vicious circle intensifies the situation as the regulation of immune response is lost and the “cytokine storm” is further activated, leading to serious consequences. Similarly, this putative “cytokine burst” pathology associated with coronaviruses is also supported by experimental SARS-CoV models, one of which showed that the severity of ALI was accompanied by an elevated expression of inflammation-related genes rather than increased viral titers. In another case, the ablation of IFN-α/β receptors or the depletion of inflammatory monocytes/macrophages caused a marked rise in the survival rate of coronavirus hosts without a change in viral load [15,16]. Both situations suggest a potential amplifying mechanism involved in CoV-induced ALI/ARDS regardless of the viral load. If a similar pathology also exists in COVID-19, the attenuation of the “cytokine storm” by targeting several key steps in the process could bring about improved outcomes. Herein, melatonin is not taken as a typical viricidal agent but it indirectly exerts anti-viral actions based on its well-reported anti-oxidative, anti-inflammatory, and immune system-enhancing properties [17,18,19,20,21,22] and, therefore, it may be useful to examine its potential effects in suppressing COVID-19 infections (see Figure 1).

## 3. Melatonin and Its Anti-Inflammatory and Anti-Oxidative Properties

Melatonin (*N*-acetyl-5-methoxytryptamine) is a multifunctional molecule with the structure of methoxyindole. It is present in almost all biological systems, in both plants and in animals. With regard to its bioactivity, it regulates circadian and seasonal biorhythms in vertebrates. Synthesis of melatonin is continuous; however, the peak of its production and release from the pineal gland takes place only at night. In adults, approximately 30 μg of melatonin are estimated to be synthesized per day, and the maximal concentration in the blood is reached in the mid-dark period. Melatonin, released from the pineal gland, is discharged into the cerebrospinal fluid and into the blood and is rapidly degraded in the liver. Melatonin has been successfully used to treat sleep disorders, atherosclerosis, respiratory diseases, and viral infections [17]. Melatonin is well known to possess potent anti-inflammatory capacities and acts via various pathways in terms of inflammatory diseases, including Sirtuin 1 (SIRT1), for the attenuation of lung injury and inflammation [23]. Similarly, melatonin suppresses nuclear factor kappa B (NF-κB) activation in ARDS, and down-regulates NF-κB activation in T cells and lung tissue [24,25,26]. NF-κB is a major transcription factor involved in the production of cytokines. Moreover, melatonin induces the nuclear translocation of NF-E2-related factor 2 (Nrf2), mediating activation of anti-oxidative phase II enzymes [19] crucial in protecting the lungs from injury. There is no clear evidence for the role of Nrf2 itself in CoV-induced ALI but the close interactions of SIRT1, NF-κB, and Nrf2 indicate their involvement in CoV-induced ALI/ARDS.

Many reports have confirmed the anti-inflammatory action of melatonin. Inflammation is known to be associated with an elevated production of cytokines and chemokines, while melatonin induces a significant reduction in pro-inflammatory cytokines [4,27]. Some of these actions are certainly mediated by melatonin membrane receptors, such as MT1 and MT2. Considering the receptor affinities, only low doses of this substance would be required; however, highly elevated doses reaching several hundred milligrams per day promote melatonin’s receptor-independent antioxidant properties. Melatonin effectively scavenges a wide range of reactive oxygen/nitrogen species (ROS/RNS), including hydroxyl radicals and the commonly overlooked carbonate radical [21,28,29,30]. Among several possibilities of formation, its mitochondrial generation may be the most important one. Under conditions of reduced gas exchange, the organism tries to enhance the arterial blood supply by producing the relaxant nitric oxide (NO) at higher rates. At the same time, the hypoxic condition can impair the mitochondrial electron flux and cause electron dissipation, which results in superoxide formation. In the presence of high NO concentrations, superoxide combines with NO to form peroxynitrite (OONO^−^). Interestingly, melatonin not only scavenges this oxidant, but also reduces its formation, by improving the mitochondrial electron flux and, thereby, decreasing superoxide generation. Generally, the protection of mitochondria by melatonin includes the prevention of the electron transport chain that causes enhanced free radical formation, control over the duration of permeability transition pore opening, and the maintenance of mitochondrial equilibrium redox balance to support mitochondrial integrity, which all have primary relevance to the return to a healthy state [31,32,33,34,35,36], in particular with regard to respiratory diseases, including the severe forms of COVID-19.

These anti-inflammatory and antioxidant properties of melatonin are also of substantial interest in pulmonary functioning under intensive care conditions. The artificial ventilation of patients bears the problem of causing undue mechanical stress to the lungs. Namely, ventilator-induced lung injury has been shown to initiate oxidative stress and inflammation. For instance, in a murine model, melatonin increased the level of the anti-inflammatory IL-10, along with improved oxygenation and reduced histological damage to the lungs [37]. Furthermore, a recent study using ramelteon, the melatoninergic agonist, in lung-injured rats revealed strong reductions in oxidative markers, reduced edema, neutrophil infiltration, the induction of apoptosis, decreased NF-κB activation and iNOS expression, and lower levels of TNFα, IL-1β, and IL-6 [38]. It should also be pointed out that the practical problem in patients with severe COVID-19 concerns the reduction of pulmonary gas exchange due to surfactant impairments by lipid peroxidation caused by infiltrating neutrophils. In vitro experiments have shown that melatonin can associate with surfactant lipids [39] and also reduce their peroxidation [40].

The anti-inflammatory actions of melatonin are also, in part, associated with mitochondrial functions, as recently outlined in the context of COVID-19 [41]. The protective mechanisms by which melatonin acts, especially under conditions of high-grade inflammation and in aging, have been reviewed [4,5]. Melatonin, in addition to its anti-inflammatory capacities, functions as an “anti-oxidative shield”, activating anti-oxidative enzymes such as catalase, superoxide dismutase, glutathione peroxidase, and phase-2 antioxidant enzymes while, on the other hand, down-regulating pro-oxidative enzymes such as nitric oxide synthase [19,42]. Viral infections and their metabolism are major sources of oxidizing agents and the anti-oxidative actions of melatonin have been documented in ALI caused by sepsis or ischemia reperfusion [43,44]. Furthermore, in advanced ALI/ARDS patients who display severe inflammation, hypoxemia, and ventilation problems, Sarma and Ward [45] noticed elevated concentrations of oxygen leading to massive oxidant generation. Regarding the beneficial roles of melatonin in COVID-19 treatment, this indoleamine was successfully applied in infants with respiratory disease [46,47] and in advanced COVID-19 pneumonia patients [8], confirming its anti-oxidative and anti-inflammatory actions in the lung. An overview of the most important functions of melatonin is provided in Figure 2.

## 4. Melatonin and Immunomodulation

From the moment the virus is inhaled and infects the epithelial cells of the respiratory tract, dendritic cells phagocytose the virus and present antigens to T cells. The resultant effector T cells function by killing the infected epithelial cells, and cytotoxic CD8+ T cells produce and release pro-inflammatory cytokines, which induce cell apoptosis [48]. Both the pathogen (CoV) and cell apoptosis trigger and amplify the immune response. Here, melatonin exerts many of its physiological actions by acting through membrane-bound MT1 and MT2 receptors, which belong to the superfamily of G-protein-coupled receptors containing the typical seven transmembrane domains and account for several of its immunological actions [49]. For instance, a resultant decrease in cyclic adenosine monophosphate (cAMP) concentration is observed upon the action of melatonin or the melatonin-mediated inhibition of cellular and humoral immune responses in mice [50]. This shows that, in animals and humans, melatonin affects both the cellular and humoral arms of the immune response [51,52]. The clinical characteristics of COVID-19 present serious disturbances in neutrophils, lymphocytes, and CD8+ T cells in peripheral blood [7,53] and melatonin exerts regulatory actions on the immune system and directly enhances the immune response by improving the proliferation and maturation of natural killer cells, T and B lymphocytes, granulocytes, and monocytes in both bone marrow and other tissues [54]. The inflammasome NLRP3 is correlated with lung diseases caused by infection, including influenza A virus and bacteria [55,56]. Since it is part of the innate immune response during lung infection, COVID-19 triggers NLRP3 activation to amplify the inflammation. Knowing the anti-inflammatory capacity of melatonin, we urge the rational use of this substance for ALI/ARDS-mediated symptoms. Indeed, the melatonin-controlled regulation of NLRP3 was shown in radiation-induced lung injury or respiratory disturbances, where melatonin distinctly reduced the infiltration of macrophages and neutrophils into the lung by inhibition of the NLRP3 inflammasome [26,57,58,59].

## 5. Melatonin and Its Adjuvant Effects

As usual, drug interactions must be considered since they may limit the use of the drugs in practice. In consideration of the common beneficial action of melatonin and attenuated metabolic processes caused by viral infections, there are reasonable imperatives to propose that this indoleamine may limit symptoms associated with viral infections, including COVID-19. Currently, it is known that severe inflammation induces multiple perturbations, such as enhanced endothelial cell apoptosis or elevation of the production of vascular endothelial growth factor (VEGF), which contributes to edema and the massive release of immune cells, while melatonin is an effective suppressor of VEGF in vascular endothelial cells [60]. Moreover, melatonin was found to be an ameliorating agent against sepsis-induced cardiomyopathy [61,62]; this effect may be also beneficial for some COVID-19 patients in whom an increased risk of sepsis and cardiac arrest accompany severe ALI/ARDS development. In addition to its effects in the lungs, melatonin is also beneficial for patients with myocardial infarction, cardiomyopathy, hypertensive heart diseases, and pulmonary hypertension [63]. Moreover, melatonin exerts neurological protection by reducing the cerebral inflammatory response, cerebral edema, and blood–brain barrier permeability [64]. Furthermore, melatonin improves sleep quality in ICU patients [65] in whom deep sedation is associated with increased long-term mortality, and the application of melatonin reduces sedation use and the frequency of pain, agitation, and anxiety [66,67]. Thus, the advantages for the use of melatonin in COVID-19 patients not only focus on the attenuation of the viral-induced respiratory disorders, but also on the overall health improvement and prevention of patients’ potential complications and their well-being.

Currently, accumulated evidence indicates that increased blood coagulation has a negative relationship with the symptoms of COVID-19 and anticoagulants are recommended to reduce the severity of COVID-19 symptoms in patients [68]. Melatonin exhibits anticoagulating activity and has been suggested to treat Ebola virus infection [69]. From this perspective, it should not be a problem to use melatonin with other anticoagulants in COVID-19 treatment. One of the main advantageous properties of melatonin is its short T_1/2_ (52.8 ± 18.1 min) [70] If physicians identify a bleeding tendency in patients and if this bleeding tendency is related to melatonin, the withdrawal of melatonin will achieve rapid (short T_1/2_) results without negative consequences. Therefore, the concomitant use of anticoagulants and melatonin is safe and it will not cause prolonged bleeding problems after its withdrawal.

Some drugs have already been suggested for the prevention and treatment of COVID-19, including chloroquine or hydroxychloroquine. However, some recent studies show that hydroxychloroquine is ineffective. Therefore, caution is suggested regarding its use. It is noteworthy that it was reported that the anti-malaria effectiveness of chloroquine was greatly increased by a melatonin antagonist, luzindole, and/or bright light at night, which reduced melatonin production. Simultaneously, even the research on the anti-malaria effect of melatonin antagonists reported that high doses of melatonin are beneficial for malaria treatment because they inhibit programmed cell death and oxidative stress [71]. Thus, applying melatonin as an adjuvant to chloroquine and hydroxychloroquine treatments of COVID-19 may reduce the necessary doses, and thus the toxicity, of these agents [72]. In addition to the drugs mentioned above, methylprednisolone is used to relieve edema, which is justified in the case of SARS, where, as previously indicated, edema contributes significantly to lung dysfunction, leading to lung failure. The activity of melatonin as a protective drug compared to methylprednisolone was studied in mice with spinal cord injuries [73]. It was shown that the protective properties of melatonin were greater than those of the steroid. The combination of these drugs has led to even greater efficacy for relieving edema [74], so melatonin can be used in combination with prednisone to relieve edema with greater efficacy in patients suffering from pneumonia with SARS-CoV-2. Finally, ribavirin, remdesivir, and other nucleotide analogs targeting RNA-dependent RNA polymerase are a popular strategy. Indeed, neither humans nor animals have the polymerase enzyme, thus, in principle, substances of this group can be highly selective. Combining nucleotide analogs with melatonin may provide additional benefits. For example, melatonin increased ribavirin potency as an anti-influenza agent, probably due to the immunomodulatory functions of melatonin. In vitro studies have shown that ribavirin in combination with melatonin shows improved properties regarding the replication inhibition of respiratory syncytial virus [75].

## 6. Melatonin and Its Safety

Melatonin, in its original form, is a “human-friendly agent”. Currently available products contain synthetic melatonin, which is structurally identical to that produced in the body. Melatonin is an endogenously synthesized molecule in the pineal gland and is present in almost all biological systems, including animals, plants, and microbes [33,76,77,78,79]. In addition to the documented anti-inflammatory benefits of melatonin, it has a very high safety profile even when used in high doses; there is no evidence that melatonin exaggerates inflammatory responses. In a randomized trial, oral intake of 25 mg/day melatonin for 6 months promoted a significant reduction in serum concentrations of IL-6 and IL-1β [80]. Similarly, in the acute phase of inflammation, including brain reperfusion [81], and coronary artery reperfusion [82], melatonin intake of 6 mg/day and 5 mg/day, respectively, for less than 5 days reduced levels of pro-inflammatory cytokines. Even doses of 1 g/day given for a month had no adverse effects in humans [83]. Additionally, Weishaupt et al. [84] treated severely affected ALS patients using 300 mg melatonin daily for 2 years, without any adverse effects. Furthermore, in acute cases after surgery, melatonin doses up to 50 mg/kg in patients showed no serious side effects [85]. The destructive inflammation and massive pathological alterations occurring in severe COVID-19 patients require adequate measures that are not satisfied by the so-called physiological levels of melatonin. For instance, the dose selected by Huang et al. [75] to treat the H1N1 virus-associated deadly influenza was inadequate. Melatonin, at a dose of 10 mg/kg/day (20 mg/kg/48 h), had a demonstrable but only slight effect, whereas a dose of 100 mg/kg/day (200 mg/kg/48 h) substantially reduced the mortality. If we convert this murine dose to the human dose according to standard dose translation, based on dividing the surface area by a factor 12.3 (120), the calculated equivalent human dose is 8.1 mg/kg/day (100/12.3 = 8.1). This dose is very similar to the dose used in two neonatal septic trials (8.1 and 8.2 mg/kg/day), as described previously [86,87]. Importantly, this dose would not cause obvious adverse effects, based on the outcomes of these clinical trials. Thus, the estimated dose to treat deadly viral infectious diseases, including COVID-19, is around 8 mg/kg/day. For a 75 kg individual, a daily dose of 600 mg may be warranted. All data indicate that large doses of melatonin, whether given chronically or for acute treatment do not cause intolerable or uncontrollable side effects and that the safety margin of melatonin for humans is as high as 3750 mg/day for a 75 kg individual [85]. Despite the high safety profile of melatonin, as summarized above, its actions in COVID-19 patients should be prudently screened for efficacy and safety.

## 7. Conclusions

Melatonin shows no harmful adverse effects in humans. Given its proven beneficial actions in multiple organs, it is imperative to introduce this indoleamine into clinical trials as an adjuvant treatment for COVID-19-infected patients. Its documented anti-inflammatory and anti-oxidative properties, actions shared by its precursor *N*-acetylserotonin and down-stream metabolites [88], have been repeatedly confirmed in respiratory disorders in both animals and humans. Considering the wealth of scientific evidence related to its high efficacy coupled with its proven safety, we encourage healthcare professionals to seriously test the potential role of melatonin against COVID-19 infection. This is urgent, since there is no reliable treatment for this devastating disease.

## Figures and Tables

**Figure 1 nutrients-12-02561-f001:**
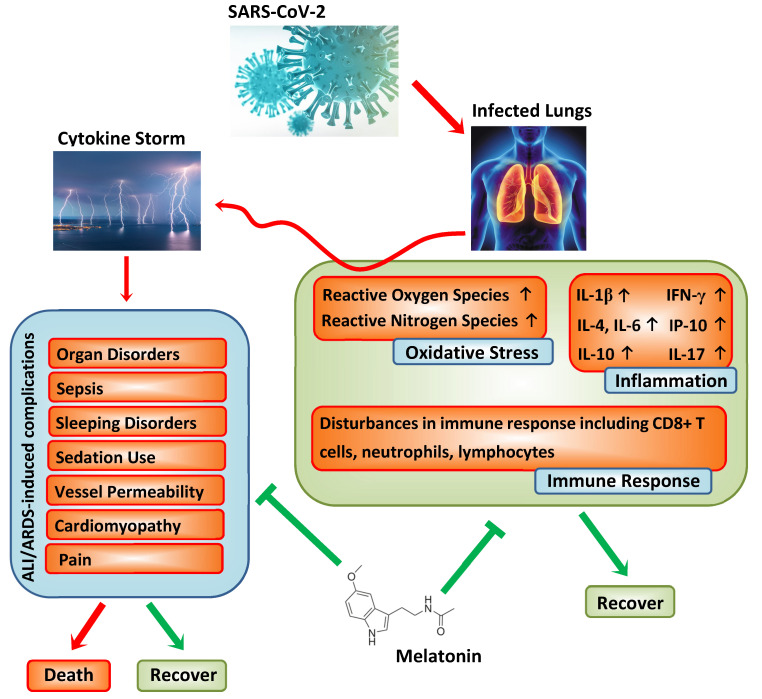
Pathogenesis of SARS-CoV-2 and adjuvant actions of melatonin. We postulate that melatonin significantly suppresses the immune response, the enhanced inflammation, and the excessive oxidative stress, triggering a “cytokine storm”. The “cytokine storm” induces acute lung injury (ALI)/acute respiratory distress syndrome (ARDS) accompanied by severe complications.

**Figure 2 nutrients-12-02561-f002:**
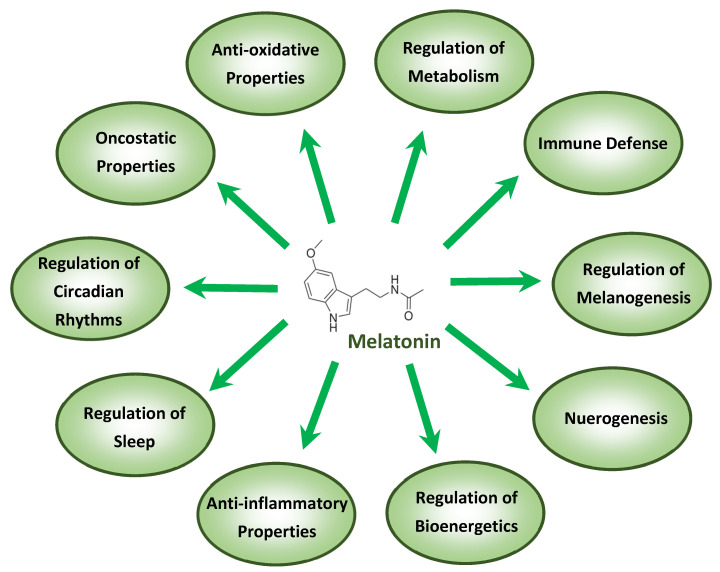
The most important functions of melatonin, some of which apply to the treatment of viral diseases including COVID-19.

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
