# Peer review of "Clinical Trials for Use of Melatonin to Fight against COVID-19 Are Urgently Needed"

_nutrients, 2020, doi:10.3390/nu12092561_

Round 1
Reviewer 1 Report
Melatonin has been used for the treatment of multi-organ diseases, including viral infections. Using it as the adjuvant therapy of COVID-19 infection that’s an interesting idea. However, important is determining the optimal dose of melatonin. So far, different doses have been used, that is from 1 mg/day to 8 mg/kg/day. Good tolerability and safety of melatonin result from its pharmacokinetic properties. In my opinion relatively high dose (about 3 – 5 mg/kg/day?) should be used in acute phase of this disease, and possible sedative effect in short-time treatment may be desired.
Conclusion: The article is well prepared in terms of content and editorial and can br published in the presented version ( except misprint in Figure 2 – look at ,,MELATONI”
Author Response
Melatonin has been used for the treatment of multi-organ diseases, including viral infections. Using it as the adjuvant therapy of COVID-19 infection that’s an interesting idea. However, important is determining the optimal dose of melatonin. So far, different doses have been used, that is from 1 mg/day to 8 mg/kg/day. Good tolerability and safety of melatonin result from its pharmacokinetic properties. In my opinion relatively high dose (about 3 – 5 mg/kg/day?) should be used in acute phase of this disease, and possible sedative effect in short-time treatment may be desired.
Conclusion: The article is well prepared in terms of content and editorial and can br published in the presented version (except misprint in Figure 2 – look at ,,MELATONI”
Authors’ response: We thank the reviewer for positive evaluation and suggestions. The misprint in Figure 2 was corrected accordingly.
Reviewer 2 Report
The review summarized the effect of melatonin in care patients, such as anti-inflammatory and anti-oxidative molecule as well as reducing patient’s vascular permeability, anxiety, sedation use, and improving their quality of sleep. The article was well organized and should be published except some small problems.
Some comments that might be helpful to the authors:
- The words are not in the same size, for example, line 45-49, line 277-279, please check the whole article carefully.
- The figure one is in a mess and many words are missing, please check and refine.
- The figure two looks funny because it is way too big.
- The melatonin in figure two was written as melatoni.
Author Response
The review summarized the effect of melatonin in care patients, such as anti-inflammatory and anti-oxidative molecule as well as reducing patient’s vascular permeability, anxiety, sedation use, and improving their quality of sleep. The article was well organized and should be published except some small problems.
Some comments that might be helpful to the authors:
- The words are not in the same size, for example, line 45-49, line 277-279, please check the whole article carefully.
- The figure one is in a mess and many words are missing, please check and refine.
- The figure two looks funny because it is way too big.
- The melatonin in figure two was written as melatoni.
Authors’ response: The authors wish to thank for all the valuable remarks and positive evaluation. Following corrections have been made as requested.
- The font size has been unified.
- Figure 1 was corrected.
- Figure 2 is corrected now.
- Misspelled writing has been corrected.
Reviewer 3 Report
This manuscript represents the author’s assertions that there is an urgent need to include melatonin as a component of clinical trials intended to discover optimal treatment plans for patients infected with the COVID-19 virus. The rationale for the inclusion of melatonin in human trials is the potential for this powerful anti-oxidant to attenuate the innate immune responses that often cause the cytokine storm and subsequent high mortality.
This manuscript is succinct and inclusive, and addresses a significant global health issue. I have only a few minor technical comments, which are itemized below. None of these are requirements; only suggestions for improvement.
1) line 24: Eliminate the first “the” in the line.
2) Check the fonts for uniformity; lines 45-66 and 277-287 are larger than the rest of the manuscript.
3) line 51: Remove the word “for”.
4) line 77: Perhaps revise to; “Although these reports were…”, or some similar re-wording to correct the syntax.
5) line 163: The sentence “In this place, somebody could point out that…” seems awkward, so could benefit by editing. (Perhaps something like: “In this [scenario, one could make the case] that the practical problem…).
6) line 170: remove the word “repeatedly”.
7) line 178: Consider revising to : “…oxygen as [a result of] the massive oxidant…”. Or: “oxygen leading to the massive oxidant…”, whichever better represents the authors meaning.
8) lines 237-246: The subject of the use of hydroxychloroquine as a potential COVID-19 treatment has become a political controversy in the United States. I feel that it would be appropriate to remark on the small number of recent studies that do not support the use of hydroxychloroquine as an effective COVID-19 treatment, so some caution is especially advisable here. Again, this is just a recommendation.
9) line 239: remove the word “apprehend” with something such as :”comprehend” or “consider”, or ‘anticipate”.
10) line 263: insert the word “to” here: “…identical [to] that produced…”
11) line 306: Change the sentence to “regarding [the] biological [significance] of…”
12) Both figures suffer from formatting issues. Figure 1 is especially in need of changing the font style or size to fit within the designated areas. Also, I am concerned whether the inclusion of any the three images at the top of the figure might constitute a copyright infringement. I have certainly seen the lung image many times on Google image searches. Some attribution might be considered here.
13) In figure 2, the word ‘melatonin’ is truncated, so the font should be adjusted.
14) My last suggestion is that the authors consider compartmentalizing some topics, as to create paragraph breaks to enhance readability.
Author Response
This manuscript represents the author’s assertions that there is an urgent need to include melatonin as a component of clinical trials intended to discover optimal treatment plans for patients infected with the COVID-19 virus. The rationale for the inclusion of melatonin in human trials is the potential for this powerful anti-oxidant to attenuate the innate immune responses that often cause the cytokine storm and subsequent high mortality.
This manuscript is succinct and inclusive, and addresses a significant global health issue. I have only a few minor technical comments, which are itemized below. None of these are requirements; only suggestions for improvement.
1) line 24: Eliminate the first “the” in the line.
2) Check the fonts for uniformity; lines 45-66 and 277-287 are larger than the rest of the manuscript.
3) line 51: Remove the word “for”.
4) line 77: Perhaps revise to; “Although these reports were…”, or some similar re-wording to correct the syntax.
5) line 163: The sentence “In this place, somebody could point out that…” seems awkward, so could benefit by editing. (Perhaps something like: “In this [scenario, one could make the case] that the practical problem…).
6) line 170: remove the word “repeatedly”.
7) line 178: Consider revising to : “…oxygen as [a result of] the massive oxidant…”. Or: “oxygen leading to the massive oxidant…”, whichever better represents the authors meaning.
8) lines 237-246: The subject of the use of hydroxychloroquine as a potential COVID-19 treatment has become a political controversy in the United States. I feel that it would be appropriate to remark on the small number of recent studies that do not support the use of hydroxychloroquine as an effective COVID-19 treatment, so some caution is especially advisable here. Again, this is just a recommendation.
9) line 239: remove the word “apprehend” with something such as :”comprehend” or “consider”, or ‘anticipate”.
10) line 263: insert the word “to” here: “…identical [to] that produced…”
11) line 306: Change the sentence to “regarding [the] biological [significance] of…”
12) Both figures suffer from formatting issues. Figure 1 is especially in need of changing the font style or size to fit within the designated areas. Also, I am concerned whether the inclusion of any the three images at the top of the figure might constitute a copyright infringement. I have certainly seen the lung image many times on Google image searches. Some attribution might be considered here.
13) In figure 2, the word ‘melatonin’ is truncated, so the font should be adjusted.
14) My last suggestion is that the authors consider compartmentalizing some topics, as to create paragraph breaks to enhance readability
Authors’ response: We thank the reviewer for positive evaluation and suggestions pointed out above (Points: 1-14).
This manuscript is a resubmission of an earlier submission. The following is a list of the peer review reports and author responses from that submission.